# Directional diffusion models for graph representation learning

**Run Yang**
SUFE and Baidu
luckyyangrun@163.sufe.edu.cn

**Yuling Yang**
SUFE
sibyllayang@163.sufe.edu.cn

**Fan Zhou**[*]
SUFE
zhoufan@mail.shufe.edu.cn

**Qiang Sun**[*]
University of Toronto
qiang.sun@utoronto.ca

## Abstract

Diffusion models have achieved remarkable success in diverse domains such as image synthesis, super-resolution, and 3D molecule generation. Surprisingly, the application of diffusion models in graph learning has garnered little attention. In this paper, we aim to bridge this gap by exploring the use of diffusion models for unsupervised graph representation learning. Our investigation commences with the identification of anisotropic structures within graphs and the recognition of a crucial limitation in the vanilla forward diffusion process when dealing with these anisotropic structures. The original forward diffusion process continually adds isotropic Gaussian noise to the data, which may excessively dilute anisotropic signals, leading to rapid signal-to-noise conversion. This rapid conversion poses challenges for training denoising neural networks and obstructs the acquisition of semantically meaningful representations during the reverse process. To overcome this challenge, we introduce a novel class of models termed *directional diffusion models*. These models adopt data-dependent, anisotropic, and directional noises in the forward diffusion process. In order to assess the effectiveness of our proposed models, we conduct extensive experiments on 12 publicly available datasets, with a particular focus on two distinct graph representation learning tasks. The experimental results unequivocally establish the superiority of our models over state-of-the-art baselines, underscoring their effectiveness in capturing meaningful graph representations. Our research not only sheds light on the intricacies of the forward process in diffusion models but also underscores the vast potential of these models in addressing a wide spectrum of graph-related tasks. Our code is available at https://github.com/statsle/DDM.

## 1   Introduction

Unsupervised representation learning through diffusion models has emerged as a prominent area of research in computer vision. Several methods that leverage diffusion models, such as those proposed by Zhang et al. (2022); Preechakul et al. (2022); Abstreiter et al. (2021); Baranchuk et al. (2021), have been put forth for representation learning. Notably, Baranchuk et al. (2021) showed that the intermediate actiations obtained from denoising networks contain valuable semantic information. Their findings emphasize the effectiveness of diffusion models in learning meaningful

---

[*]Corresponding authors.

37th Conference on Neural Information Processing Systems (NeurIPS 2023).

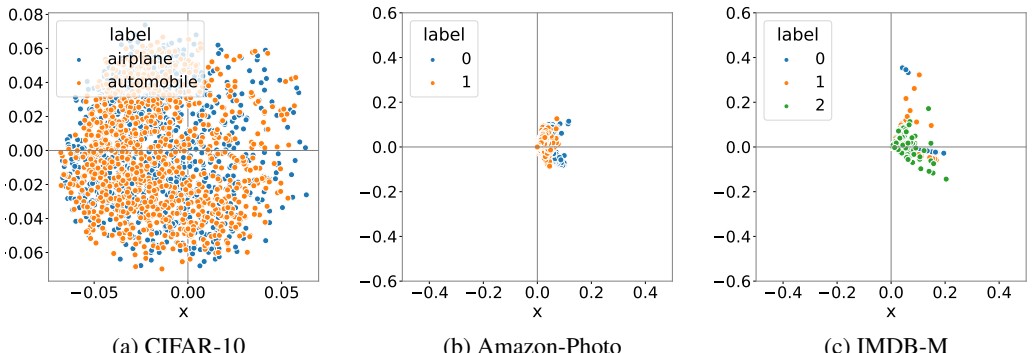

Figure 1: 2D visualization of the data using SVD decomposition. (a) Visualization of the node features in CIFAR-10, with different colors indicating different labels. (b) Visualization of the two classes in Amazon-Photo. (c) Visualization of the graph features in IMDB-M.

visual representations. More recently, Choi et al. (2022) showed that restoring data corrupted with certain noise levels offers a proper pretext task for the model to learn rich visual concepts.

Despite the increasing research on diffusion models in computer vision, there is still a noticeable shortage of studies exploring the use of diffusion models in graph learning. Previous works, such as those by Haefeli et al. (2022) and Jo et al. (2022), have primarily concentrated on employing diffusion models for generating discrete graph structures. However, the realm of graph representation learning, a fundamental and challenging aspect of graph learning, has yet to harness the potential of diffusion models. To effectively adapt and incorporate diffusion models into graph representation learning, and to facilitate progress in this domain, it is imperative to identify and understand the obstacles hindering the application of diffusion models.

To gain insights into the limitations of the vanilla diffusion models which are commonly adopted for image generation tasks (Ho et al., 2020), we first investigate the underlying structural disparities between images and graphs. In particular, we employ singular value decomposition (SVD) on both image and graph data, and visualize the projected data in a 2-dimensional plane, as shown in Figure 1. This figure illustrates that the projected data from Amazon-Photo and IMDB-M exhibit strong anisotropic structures along only a few directions, while the projected images from CIFAR-10 form a relatively more isotropic distribution within a circular shape centered around the origin. This observation strongly implies that graph data may feature distinctive anisotropic and directional structures that are less prevalent in image data. As we will subsequently demonstrate, the vanilla diffusion models with isotropic forward diffusion processes result in a rapid decrease in the signal-to-noise ratios (SNRs), which in turn diminishes their effectiveness in learning anisotropic structures. Therefore, it is imperative to develop new approaches that can effectively account for these anisotropic structures.

This paper presents directional diffusion models as a solution for better learning anisotropic structures. Our approach involves incorporating data-dependent and directional noise into the forward diffusion process, effectively mitigating the challenge of rapid signal-to-noise ratio deterioration. The intermediate activations obtained from the denoising network excel at capturing valuable semantic and topological information crucial for downstream tasks. Consequently, the proposed directional diffusion models offer a promising approach for generative graph representation learning. Numerically, we perform experiments on 12 benchmark datasets covering both node and graph classification tasks. The results consistently highlight the superior performance of our models when compared to state-of-the-art contrastive learning and generative approaches (Hou et al., 2022). Notably, for graph classification problems, our directional diffusion models even surpass supervised baseline methods, underscoring the considerable potential of diffusion models in the field of graph representation learning.

Our main contributions are summarized as follows.

1. We contribute to the exploration of anisotropic structures in graph data, being among the pioneers in the literature. We demonstrate that the vanilla forward diffusion process with isotropic white noise leads to a rapid decline in signal-to-noise ratios for graph learning problems. This issue hampers the ability of denoising networks to extract fine-grained feature representations across a wide range of SNRs.

2. We propose novel directional diffusion models specifically designed for graph data, incorporating data-dependent and directional noise in the forward diffusion process. Our proposed models effectively address the issue of the rapid decline of SNRs, enabling better graph representation learning.

3. Numerically, our proposed directional diffusion models outperform state-of-the-art self-supervised methods and even supervised methods on 12 benchmark datasets. Additionally, we provide comprehensive ablation studies to gain a deeper understanding of the mechanisms underlying directional diffusion models.

## 2 Related work

**Graph representation learning**   Graph representation learning aims to embed nodes or entire graphs into a low-dimensional vector space, where the structural and relational properties can be used for downstream tasks. Two prevalent paradigms for graph representation learning are contrastive learning and generative self-supervised learning. Contrastive learning approaches such as DGI (Velickovic et al., 2019), Infograph (Sun et al., 2019), GraphCL (You et al., 2020), GRACE (Zhu et al., 2020), and GCC (Qiu et al., 2020), have achieved promising results in some particular graph learning tasks. These methods leverage local-global mutual information maximization for node and graph representation learning. GraphCL learns node embeddings that are invariant to graph-level transformations, while GRACE and GCC use subgraph sampling and graph perturbation to create augmented pairs. Generative self-supervised learning aims to recover masked components of the input data through approaches such as GraphMAE (Hou et al., 2022), a masked graph autoencoder that focuses on feature reconstruction by utilizing a masking strategy and scaled cosine error. This method outperforms self-supervised learning baselines, and revitalizes the concept of generative self-supervised learning on graphs. GPT-GNN (Hu et al., 2020b) is a recent approach that introduces a self-supervised attributed graph generation task to pre-train a GNN so that it can capture the structural and semantic properties of the graph.

**Denoising diffusion probabilistic models**   Denoising diffusion probabilistic models (Ho et al., 2020; Song et al., 2020), or simply diffusion models, are a class of probabilistic generative models that turn noise to a data sample and thus are mainly used for generation tasks (Dhariwal and Nichol, 2021; Rombach et al., 2022).Recently, diffusion models have been used as a representation learning toolbox in computer vision (Preechakul et al., 2022; Abstreiter et al., 2021; Baranchuk et al., 2021). For instance, Preechakul et al. (2022) proposed Diff-AE, a method that concurrently trains an encoder to discover high-level semantics and a conditional diffusion model that uses these representations as input conditions. Abstreiter et al. (2021) augmented the denoising score matching framework to enable representation learning without any supervised signal. Recently, diffusion models have also found applications in handling graph data. Haefeli et al. (2022) showed that diffusion models for graphs benefited from discrete state spaces. Jo et al. (2022) proposed the graph diffusion using stochastic differential equations. To the best of our knowledge, there have been no works for diffusion-model-based graph representation learning.

## 3 The effect of anisotropic structures

As discussed in the introduction section, significant structural differences exist between graphs and natural images. In addition to the analysis of the Amazon-Photo and IMDB-M datasets, we conducted similar examinations on all other graph benchmark datasets, and the detailed results can be found in the appendix. Moreover, it is important to highlight that the anisotropic structures, referred to as categorical directional dependence, are frequently encountered in natural language data (Gao et al., 2019; Li et al., 2020). It is intriguing to observe that diffusion models have not yet made significant strides in the realm of natural language processing. This further underscores the significance of

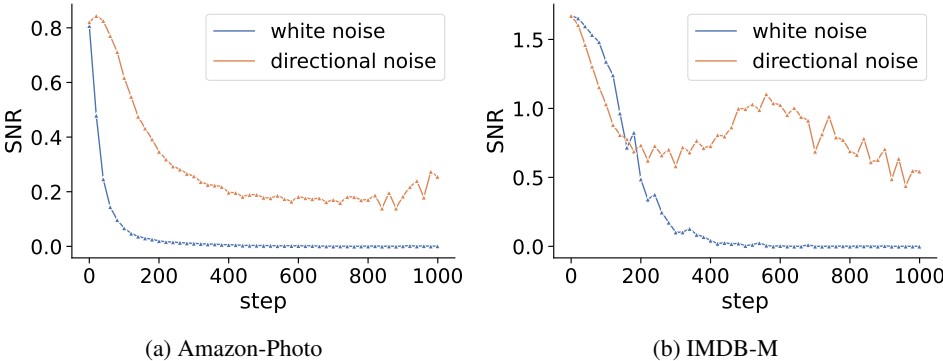

|       | (a) Amazon-Photo | (b) IMDB-M |
|-------|------------------|------------|

Figure 2: The signal-to-noise ratio curve along different diffusion steps.

investigating and tackling the challenges posed by anisotropic structures within the framework of diffusion models.

This section delves deeper into the examination of how anisotropic structures in graphs impact the effectiveness of vanilla diffusion models. In the vanilla forward diffusion process, isotropic Gaussian noise is sequentially added to the raw data point $x_0 \sim q(x_0)$ until it transforms into isotropic white noise $\mathcal{N}(0, \mathbf{I})^2$. This approach is reasonable when the data follow isotropic distributions, as it gradually transforms the data into noise, generating a sequence of noisy samples with a wide range of SNRs. However, in cases where the data exhibits anisotropy, the addition of isotropic noise can quickly contaminate the data structure, causing the SNRs to rapidly approach zero. Consequently, denoising networks struggle to extract fine-grained feature representations.

To explore the impact of introducing isotropic noise on the learning of anisotropic graphs, we conduct an experiment to measure the SNRs for both node and graph classification tasks at each forward step. We observe how these SNRs change throughout the forward diffusion process. Initially, we pre-train a graph neural network (GNN) denoted as $\mathbf{E}$ to serve as a feature extractor, projecting the graph data into a linearly separable space. Subsequently, we optimize the weight vector $w \in \mathbb{R}^d$ in this linear space using Fisher's linear discriminant analysis. The weight vector $w$ is then employed to compute the SNR at each forward diffusion step, where $\mathrm{SNR} = w^\mathsf{T} S_B w / w^\mathsf{T} S_W w$. In this equation, $S_B$ represents the scatter between class variability, and $S_W$ represents the scatter within-class variability. This SNR quantifies the discriminative power of the learned representations at different stages of the diffusion process.

We conducted this experiment on all graph benchmark datasets to assess the impact of isotropic noise on learning anisotropic graphs. Here, we present the results for IMDB-M and Amazon-Photo in Figure 2a, while the additional results can be found in the appendix. In Figure 2a, we observe that for anisotropic graph data and isotropic noise, the SNR rapidly decreases to 0 at around 50 steps for Amazon-Photo and 400 steps for IMDB-M. Furthermore, the SNR remains close to 0 thereafter, indicating that the incremental isotropic white noise quickly obscures the underlying anisotropic structures or signals. Consequently, the denoising networks are unable to learn meaningful and discriminative feature representations that can be effectively utilized for downstream classification tasks. In contrast, when utilizing our directional diffusion models, which incorporate a data-dependent and directional forward diffusion process (to be introduced later), the SNR declines at a slower pace. This slower decline enables the extraction of fine-grained feature representations with varying SNRs, preserving the essential information of the anisotropic structures.

Overall, these studies underscore the significance of considering anisotropic data structures when designing forward diffusion processes and the corresponding diffusion models, especially in the context of graph data.

---

[2]Bold symbols are used for matrices, but not for vectors.

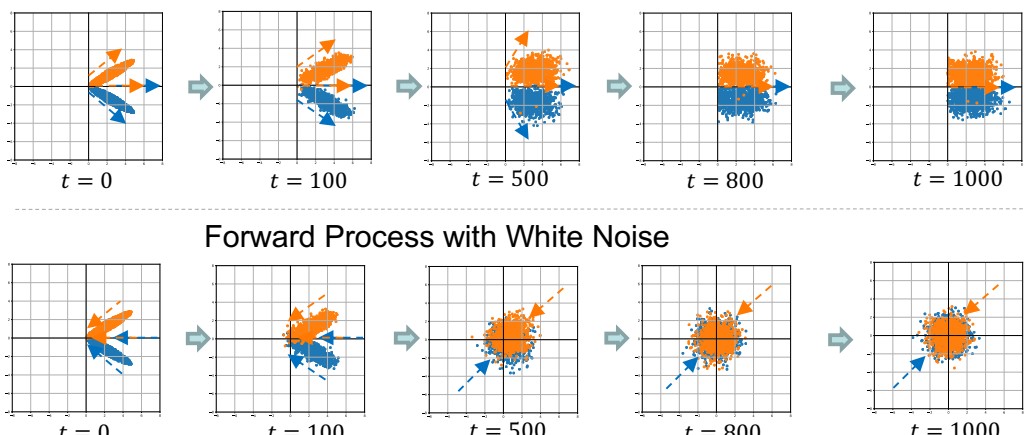

Figure 3: Directional noise vs white noise. We sample directional and white noise[1], and subsequently add the noise in a sequential manner. The upper panel displays the samples with directional noise at the diffusion steps $t = 0, 100, 500, 800, 1000$, while the lower panel exhibits the samples with white noise at these identical diffusion steps. The two distinct colors indicate two different classes.

## 4 Directional diffusion models

In this section, we start by introducing the requisite notation. Subsequently, we propose the directional diffusion models (DDMs), with a specific focus on their application to graph representation learning. Furthermore, we discuss how to extract feature representations from DDMs, a pivotal aspect for downstream tasks.

**Notation** We denote a graph by $G = (\mathbf{V}, \mathbf{A}, \mathbf{X})$, where $\mathbf{V}$ is the node set, $N = |\mathbf{V}|$ is the node number, $\mathbf{A} \in \mathbb{R}^{N \times N}$ is the adjacency matrix (binary or weighted), and $\mathbf{X} = (x_1, x_2, \cdots, x_N)^{\mathrm{T}} \in \mathbb{R}^{N \times d}$ is the node feature matrix. Our goal is to learn a network, denoted as $f : \mathbb{R}^{N \times d} \times \mathbb{R}^{N \times N} \to \mathbb{R}^{N \times d_h}$, to encode graph features into representations $\mathbf{H} = (h_1, h_2, \cdots, h_N)^{\mathrm{T}} \in \mathbb{R}^{N \times d_h}$, where $h_i \in \mathbb{R}^{d_h}$ is the representation for node $i$. Mathematically, we have $\mathbf{H} = f(\mathbf{X}, \mathbf{A})$. These representations can subsequently be used for downstream tasks, such as graph and node classification.

**Directional diffusion models** In the preceding section, our investigation uncovered a pivotal factor contributing to the underwhelming performance of vanilla diffusion models in graph learning: the swift deterioration of signal-to-noise ratios. To tackle this challenge, we introduce the *directional noise* in the forward diffusion process, which involves transforming the isotropic Gaussian noise into an anisotropic noise by incorporating two additional constraints. These two constraints play a vital role in enhancing the efficacy of vanilla diffusion models.

Let $G_t = (\mathbf{A}, \mathbf{X}_t)$ be the noisy graph at the $t$-th forward diffusion step, where $\mathbf{X}_t = \{x_{t,1}, x_{t,2}, ..., x_{t,N}\}$ represents the learned features at the $t$-th step. Specifically, the node feature $x_{t,i} \in \mathcal{R}^d$ of node $i$ at time $t$ is obtained as follows:

$$x_{t,i} = \sqrt{\bar{\alpha}_t} x_{0,i} + \sqrt{1 - \bar{\alpha}_t} \epsilon', \tag{1}$$

$$\epsilon' = \mathrm{sgn}(x_{0,i}) \odot |\bar{\epsilon}|, \tag{2}$$

$$\bar{\epsilon} = \mu + \sigma \odot \epsilon \quad \text{where } \epsilon \sim \mathcal{N}(0, \mathbf{I}), \tag{3}$$

where $x_{0,i}$ is the raw feature vector of node $i$, $\mu \in \mathbb{R}^d$ and $\sigma \in \mathbb{R}^d$ consist of the means and standard deviations of the $d$ features across $N$ nodes, respectively. The symbol $\odot$ denotes the Hadamard product. During the mini-batch training, $\mu$ and $\sigma$ are calculated using graphs within the batch. The parameter $\bar{\alpha}_t := \prod_{i=0}^{t}(1 - \beta_i) \in (0, 1)$ represents the variance schedule (Ho et al., 2020) and is parameterized by a decreasing sequence $\{\beta_{1:T} \in (0, 1)\}$.

---

[1]The generation parameters are provided in the appendix.

In contrast to the conventional forward diffusion process, our directional diffusion models integrate two additional constraints, as denoted in equations (2) and (3). The second constraint, (3), transforms the data-independent Gaussian noise into an anisotropic and batch-dependent noise. In this constraint, each coordinate of the noise vector shares the same empirical mean and empirical standard deviation as the corresponding coordinate in the data within the same batch. This constraint confines the diffusion process to the local neighborhood of the batch, preventing excessive deviation from the batch and preserving local coherence.

The first constraint, (2), aligns the noise $\epsilon'$ with the feature $x_{0,i}$ to ensure that they share the same coordinate signs. This guarantees that adding noise does not result in noisy features pointing in the opposite direction of $x_{0,i}$. By preserving the directionality of the original feature, this constraint plays a crucial role in maintaining the inherent data structure during the forward diffusion process. These two constraints work in tandem to ensure that the forward diffusion process respects the underlying data structure and mitigates the rapid degradation of signals. Consequently, the SNR decays slowly, enabling our directional diffusion models to effectively extract meaningful feature representations across various steps. This, in turn, enhances the utility of these representations in downstream tasks by providing reliability and informativeness.

To illustrate the impact of directional noise, we refer to the experiments conducted in Section 3. Our newly proposed "directional noise" ensures a smoother decline of the SNRs throughout the diffusion process, confirming our initial intuition. To further visualize the differences between using directional noise and isotropic noise in the forward diffusion process, we conducted simulations on two ellipses and sequentially added noise, as depicted in Figure 3. The figure clearly illustrates the distinct behaviors exhibited by the two types of noise. With directional noise, the samples maintain a clear decision boundary, indicating the preservation of discriminative structures during the diffusion process. Conversely, samples with white noise quickly blend into pure noise, leading to the loss of discriminative information. This visual comparison unequivocally underscores the superiority of directional noise in preserving the structural information of the data during the forward diffusion process.

**Model architecture**    We follow the same training strategy as in the vanilla diffusion models, where we train a denoising network $f_\theta$ in the reverse diffusion process. Since the posterior of the forward process with directional noise cannot be expressed in a closed form, we borrow the idea from Bansal et al. (2022); Li et al. (2022) and let the denoising model $f_\theta$ directly predict $\mathbf{X}_0$. The loss function $\mathcal{L}$ is defined as the expected value of the Euclidean distance between the predicted feature representation $f_\theta(\mathbf{X}_t, \mathbf{A}, t)$ and the original feature representation $\mathbf{X}_0$:

$$\mathcal{L} = \mathbb{E}_{\mathbf{X_0}, t} \| f_\theta(\mathbf{X}_t, \mathbf{A}, t) - \mathbf{X}_0 \|^2. \tag{4}$$

This loss function ensures that the model predicts $\mathbf{X}_0$ at every step.

To parameterize the denoising network $f_\theta$, we employ a symmetrical architecture inspired by the successful UNet architecture in computer vision (Dhariwal and Nichol, 2021). Figure 4 illustrates our DDM framework, comprising four GNN layers and one multilayer perception (MLP). The first two GNN layers function as the *encoder*, responsible for denoising the target node by aggregating neighboring information. The final two GNN layers serve as the *decoder*, mapping the denoised node features to a latent code and smoothing the latent code among neighboring nodes. To address the potential issue of over-smoothing and account for long-distance dependencies in the graph, we introduce skip-connections between the encoder and decoder. The algorithm, along with the mini-batch training procedure, is presented in the appendix.

**Learning representations**    For a given graph $G = (\mathbf{A}, \mathbf{X})$, the learned node-level representations are obtained from the activations of the denoising network $f_\theta$ at user-selected time steps. It is important to note that we only utilize the activations from the decoder of $f_\theta$ since they incorporate the encoder activations through skip connections. As depicted in Figure 4, at each time step $k$, we introduce $k$ steps of directional noise following (1) and employ the denoising network $f_\theta$ to denoise and encode the noisy data $\mathbf{X}_k$. The decoder of $f_\theta$ maps the denoised node features to a latent code while smoothing the latent code among neighboring nodes. We extract the activations from the decoder of $f_\theta$ and concatenate them to obtain $\mathbf{H}_k = \{h_{k,1}, h_{k,2}, \cdots, h_{k,N}\} \in \mathbb{R}^{N \times d_h}$. The complete pipeline is presented in the appendix.

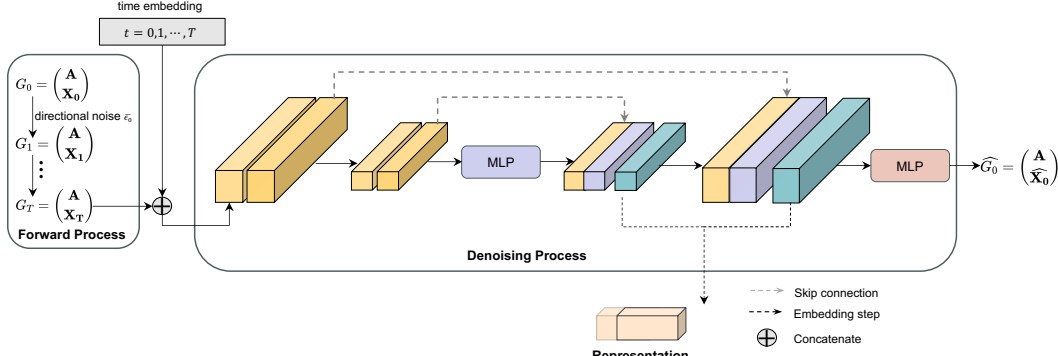

Figure 4: The pipeline of our model: (1) Adding directional noise to the original graph $G_0$. (2) Extracting the last two GNN's feature maps in the denoising network as the representation of the graph.

## 5 Experiments

This section provides an evaluation of the directional diffusion models from two perspectives. First, we compare our models with existing state-of-the-art methods on various graph learning tasks, including node and graph classification tasks. This allows us to assess the effectiveness of our approach for graph representation learning problems. Second, we conduct several studies to gain a better understanding of the effect of our directional noise and evaluate the necessity of our design choices.

In all experiments, we follow a two-step process. First, we pre-train a DDM on the dataset in an unsupervised manner. Then, we extract feature representations from diffusion steps $50, 100, 200$ using the pre-trained model. Although this approach is inspired by the experimental results and insights from Section 3, it is deliberately not fine-tuned for each dataset. Ideally, fine-tuning with carefully selected steps for each dataset could further improve the performance.

### 5.1 Graph classification

To demonstrate the effectiveness of our method, we conduct comparisons with state-of-the-art (SOTA) unsupervised learning methods, which include GCC (Qiu et al., 2020), Infograph (Sun et al., 2019), GraphCL (You et al., 2020), JOAO (You et al., 2021), MVGRL (Hassani and Khasahmadi, 2020), and GraphMAE (Hou et al., 2022). We also compare our approach with supervised learning methods, specifically GIN (Xu et al., 2018) and DiffPool (Ying et al., 2018). These experiments are conducted on seven widely-used datasets, namely MUTAG, IMDB-B, IMDB-M, PROTEINS, COLLAB, and REDDIT-B (Yanardag and Vishwanathan, 2015). Node degrees are used as initial node features for IMDB-B, IMDB-M, REDDIT-B, and COLLAB, while node labels are employed for MUTAG and PROTEINS, in accordance with prior literature (Hou et al., 2022). We extract graph-level representations at various steps, train linear SVMs using LIBSVM (Chang and Lin, 2011), and independently test them. The final predictions are obtained by majority vote, and we report the average accuracy and standard deviation, calculated after five runs. Additional details on hyper-parameters can be found in the appendix.

Table 1 presents the results, clearly indicating that our DDM achieves the most competitive, if not the best, performance across all benchmark datasets. Notably, our DDM outperforms even the supervised approaches in certain experiments, such as IMDB-B, COLLAB, and MUTAG. This remarkable performance can be attributed to two key factors. First, from a data perspective, these datasets have limited information in their node features, which can hinder the accuracy of supervised learning (Hou et al., 2022). By leveraging directional noise diffusion, our DDM effectively acts as a pseudo-infinite-step data augmentation technique, generating numerous samples while preserving the classification boundary. This augmentation significantly enhances the effectiveness of unsupervised learning. Second, from a model perspective, the DDM framework harnesses the power of directional

Table 1: Results in unsupervised representation learning for graph classification.

| Dataset | IMDB-B | IMDB-M | COLLAB | REDDIT-B | PROTEINS | MUTAG |
|---------|--------|--------|--------|----------|----------|-------|
| GIN | 75.1±5.1 | 52.3±2.8 | 80.2±1.9 | 92.4±2.5 | 76.2±2.8 | 89.4±5.6 |
| DiffPool | 72.6±3.9 | - | 78.9±2.3 | 92.1±2.6 | 75.1±2.3 | 85.0±10.3 |
| Infograph | 73.03±0.87 | 49.69±0.53 | 70.65±1.13 | 82.50±1.42 | 74.44±0.31 | 89.01±1.13 |
| GraphCL | 71.14±0.44 | 48.58±0.67 | 71.36±1.15 | 89.53±0.84 | 74.39±0.45 | 86.80±1.34 |
| JOAO | 70.21±3.08 | 49.20±0.77 | 69.50±0.36 | 85.29±1.35 | 74.55±0.41 | 87.35±1.02 |
| GCC | 72 | 49.4 | 78.9 | 89.8 | - | - |
| MVGRL | 74.20±0.70 | 51.20±0.50 | - | 84.50±0.60 | - | 89.70±1.10 |
| GraphMAE | 75.52±0.66 | 51.63±0.52 | 80.32±0.46 | 88.01±0.19 | 75.30±0.39 | 88.19±1.26 |
| DDM | **76.40±0.22** | **52.53±0.31** | **81.72±0.31** | 89.15 ±1.3 | **75.47 ±0.50** | **91.51 ±1.45** |

Table 2: Results in unsupervised representation learning for node classification.

| Dataset | Cora | Citeseer | PubMed | Ogbn-arxiv | Computer | Photo |
|---------|------|----------|--------|------------|----------|-------|
| GAT | 83.0 ± 0.7 | 72.5 ± 0.7 | 79.0 ± 0.3 | 72.10 ± 0.13 | 86.93 ± 0.29 | 92.56 ± 0.35 |
| DGI | 82.3 ± 0.6 | 71.8 ± 0.7 | 76.8 ± 0.6 | 70.34 ± 0.16 | 83.95 ± 0.47 | 91.61 ± 0.22 |
| MVGRL | 83.5 ± 0.4 | 73.3 ± 0.5 | 80.1 ± 0.7 | - | 87.52 ± 0.11 | 91.74 ± 0.07 |
| BGRL | 82.7 ± 0.6 | 71.1 ± 0.8 | 79.6 ± 0.5 | 71.64 ± 0.12 | 89.68 ± 0.31 | 92.87 ± 0.27 |
| InfoGCL | 83.5 ± 0.3 | 73.5 ± 0.4 | 79.1 ± 0.2 | - | - | - |
| CCA-SSG | 84.0 ± 0.4 | 73.1 ± 0.3 | 81.0 ± 0.4 | 71.24 ± 0.20 | 88.74 ± 0.28 | 93.14 ± 0.14 |
| GPT-GNN | 80.1 ± 1.0 | 68.4 ± 1.6 | 76.3 ± 0.8 | - | - | - |
| GraphMAE | 84.2 ± 0.4 | 73.4 ± 0.4 | 81.1 ± 0.4 | 71.75 ± 0.17 | 88.63 ± 0.17 | 93.63 ± 0.22 |
| DDM | 83.4 ± 0.2 | **74.3 ± 0.3** | **81.7 ± 0.8** | 71.29 ± 0.18 | **90.56 ± 0.21** | **95.09 ± 0.18** |

noise and ensures that the learned representations capture discriminative information by preventing the rapid decay of signal-to-noise ratios.

## 5.2 Node classification

To assess the quality of the node-level representations produced by our method, we conduct evaluations of DDM on six standard benchmark datasets: Cora, Citeseer, PubMed (Yang et al., 2016), Ogbn-arxiv (Hu et al., 2020a), Amazon-Computer (Zhang et al., 2021), and Amazon-Photo (Zhang et al., 2021). We follow the publicly available data-split schema and employ the evaluation protocol used in the literature. Graph-level representations are extracted at different diffusion steps, and an independent linear classifier is trained for each step. The final predictions are determined through majority voting, and we report the mean accuracy on the test nodes. Additional details regarding hyperparameters can be found in the appendix.

We compare DDM with state-of-the-art generative unsupervised models, specifically GPT-GNN (Hu et al., 2020b) and GraphMAE (Hou et al., 2022). Furthermore, we include the results of contrastive unsupervised models for comparison, which include DGI (Velickovic et al., 2019), MVGRL (Hassani and Khasahmadi, 2020), GRACE (Zhu et al., 2020), BGRL (Thakoor et al., 2021), InfoGCL (Xu et al., 2021), and CCA-SSG (Zhang et al., 2021). As shown in Table 2, DDM achieves competitive results across all benchmark datasets. This underscores the capacity of the generative diffusion method to learn meaningful node-level representations and highlights the effectiveness of DDM in node-level tasks. Notably, the node features used in node classification are text embeddings, showcasing the efficacy of our directional noise in continuous word vector spaces.

## 5.3 Understanding the directional noise

The aforementioned studies provide evidence that our approach either outperforms or is on par with existing state-of-the-art (SOTA) methods. To gain a deeper understanding, we conduct a comprehensive investigation into the impact of different types of noise. Additionally, we analyze the effect of directional noise by removing the two constraints specified in equations (3) and (2).

We examine the representations extracted from models trained with directional noise and isotropic white noise at each reverse step. The results, as depicted in Figure 5, reveal significant differences between the two approaches. With white noise, only the representations corresponding to the early steps of the reverse process contained useful discriminative information, while the representations for later steps are mostly uninformative. This is in stark contrast to the case of directional noise, where the learned representations consistently preserve sufficient information for downstream classification tasks.

In our additional experiments, we consistently observe that directional diffusion models outperform vanilla diffusion models across all datasets, particularly in node classification tasks. This superior performance can be attributed to the nature of node classification datasets, which often use word vectors as node features. These word vectors have higher feature dimensionality and greater anisotropy. The effectiveness of our directional approach is further supported by these findings, reinforcing its value in graph representation learning.

To examine the impact of different schedulers, we compare the performance of the vanilla diffusion model using isotropic white noise and DDM under the cosine and sigmoid scheduler in Chen (2023). Table 3 collects the results. Different schedulers indeed influence the model performance. However, the performance of the vanilla diffusion model with isotropic white noise heavily relies on the hyperparameters of the scheduler. Yet, irrespective of the scheduler employed, our proposed data-dependent anisotropic noise consistently yields superior performance.

Table 3: Results for different schedulers.

| Noise schedule function | Noise type | Citeseer | PubMed | MUTAG |
|---|---|---|---|---|
| cosine (s=0, e=1, $\tau = 1$) | DDM | 0.715 | 0.824 | 0.867 |
| cosine (s=0, e=1, $\tau = 1$) | White Noise | 0.371 | 0.453 | 0.692 |
| sigmoid (s=0, e=3, $\tau = 1$) | DDM | 0.710 | 0.806 | 0.877 |
| sigmoid (s=0, e=3, $\tau = 1$) | White Noise | 0.581 | 0.434 | 0.691 |

Lastly, we conduct an ablation study to examine the effects of the two constraints. Table 4 presents results, where "w/o R" indicates the removal of the constraint (2) and "w/o S&R" indicates the removal of both constraints. As shown in Table 4, the introduction of anisotropic Gaussian noise generated through (3) led to a significant improvement compared to isotropic Gaussian noise. Furthermore, the inclusion of constraint (2) provided an additional and indispensable improvement. This finding further confirms the importance of making the noise in the forward process data-dependent and anisotropic.

Table 4: An ablation study on the two constraints.

| Dataset | w/o S&R | w/o R | Full |
|---|---|---|---|
| Citeseer | 34.37±0.5 | 60.77±0.2 | **74.3 ± 0.3** |
| PubMed | 73.07±0.7 | 77.60±0.4 | **81.7 ± 0.8** |
| IMDB-M | 49.80±0.53 | 50.87±0.49 | **52.53±0.31** |
| COLLAB | 80.50±0.36 | 81.04±0.17 | **81.72±0.31** |
| MUTAG | 82.89±1.16 | 87.25±1.12 | **91.51±1.45** |

## 6 Conclusions

This paper unveils the presence of anisotropic structures in graphs, which pose challenges for vanilla diffusion models in graph representation learning. To address this limitation, we introduce directional diffusion models, a novel class of diffusion models that leverage data-dependent and anisotropic noise to better handle anisotropic structures. Through experiments conducted on 12 benchmark datasets, we demonstrate the effectiveness of our proposed method.

There are several promising avenues for future research. One direction is to develop methods that can automatically determine the optimal set of diffusion steps for each dataset, further enhancing the performance of our directional diffusion models. This could involve techniques such as adaptive selection of diffusion steps. Additionally, exploring the application of our method to computer vision and natural language processing tasks holds great potential for advancing these domains. By adapting and extending our directional diffusion models to these areas, we may leverage their inherent strengths to improve representations and enable effective learning tasks such as image recognition, object detection, sentiment analysis, and language understanding.

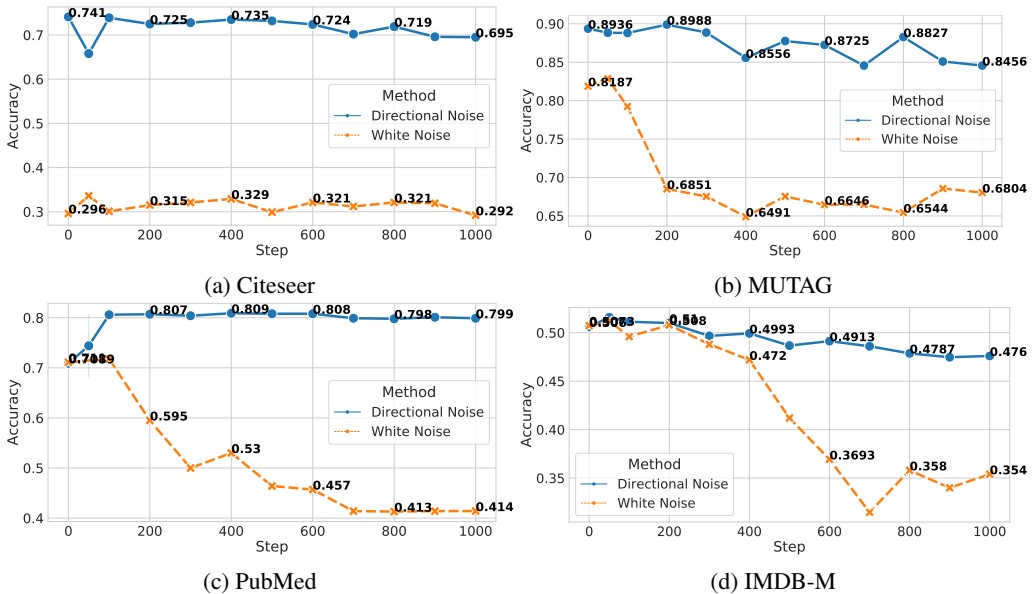

Figure 5: Comparing the accuracy in downstream tasks when using representations extracted from models trained with directional noise and white noise at every step of the reverse process.

# 7 Acknowledgements

FZ is supported by National Natural Science Foundation of China (12001356), Shanghai Education Development Foundation and Shanghai Municipal Education Commission (Chenguang Program), Zhejiang Lab (Open Research Project NO. 2022RC0AB06), Shanghai Research Center for Data Science and Decision Technology. Fz is Supported by the Fundamental Research Funds for the Central Universities: "High-Quality Development of Digital Economy: An Investigation of Characteristics and Driving Strategies (Grant Numbers 2023110139). QS is supported in part by a NSERC Discovery Grant, a New Frontiers in Research Fund, and a Data Sciences Institute Catalyst Grant.

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
