# Supplementary Material

## A  Visualizing all other graph datasets

We visualize 9 graph datasets using their SVD decomposition, as shown in Figure 6. All the existing graph data exhibits strong anisotropic data structures.

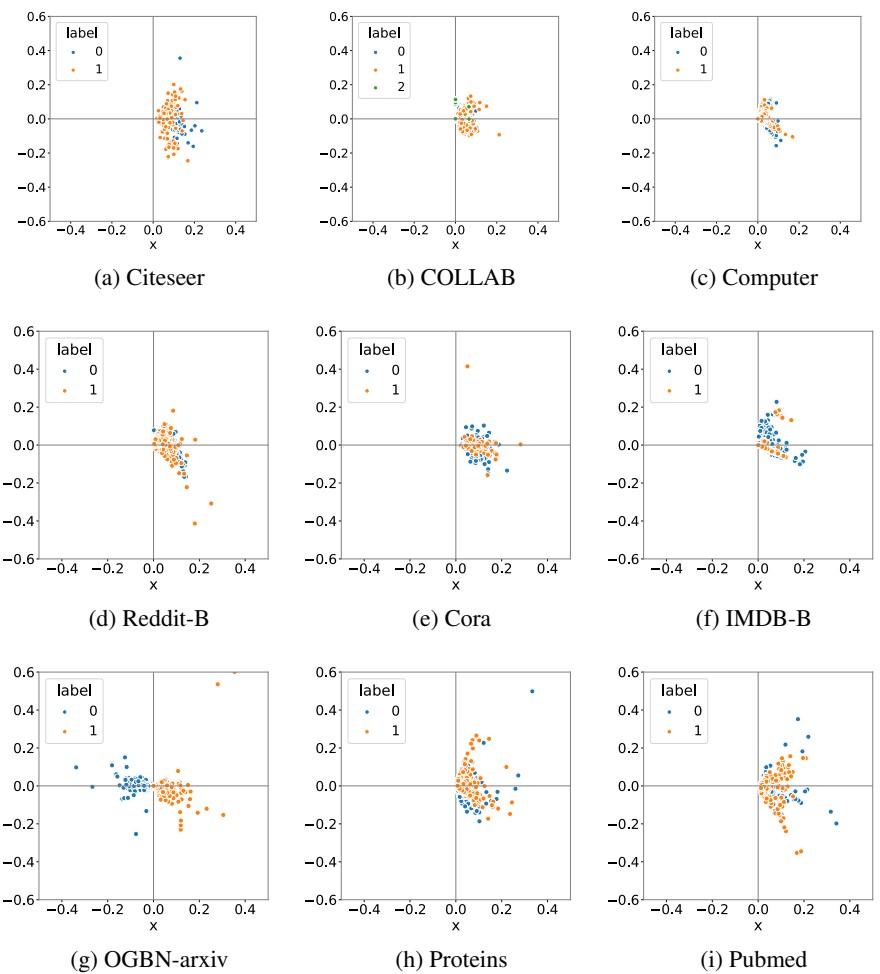

Figure 6: 2D visualization of the data using SVD decomposition.

## B    Signal-to-noise ratios analysis

In this section, we examine the signal-to-noise ratios at various diffusion steps in various graph datasets, as shown in Fig 7.

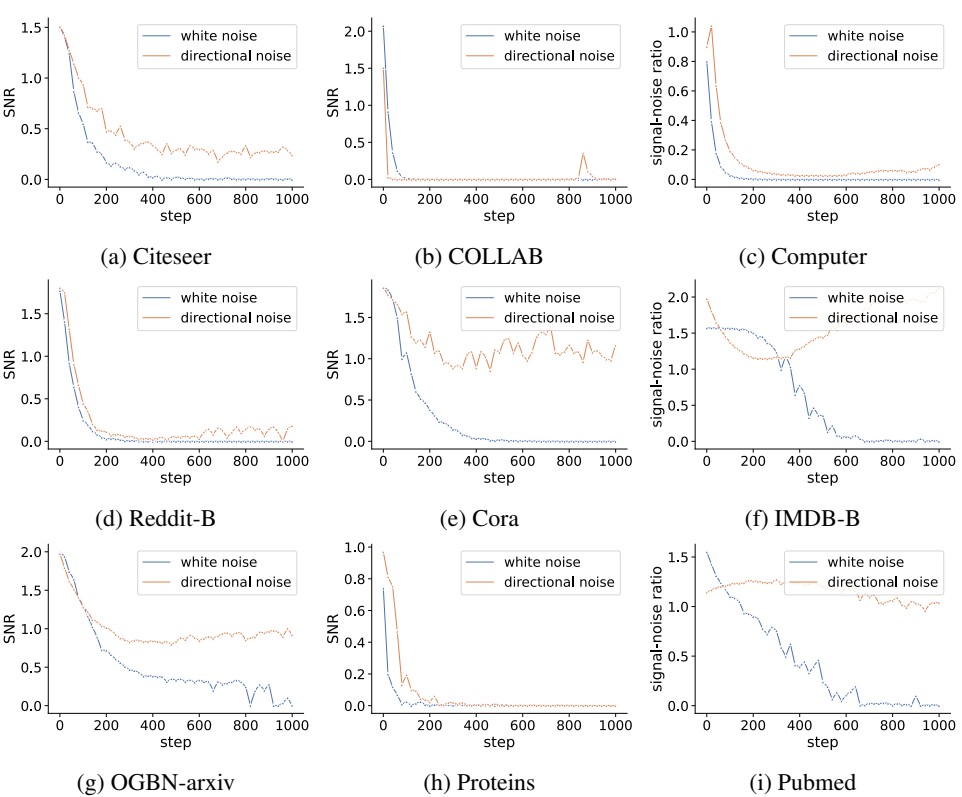

(a) Citeseer      (b) COLLAB      (c) Computer

(d) Reddit-B      (e) Cora      (f) IMDB-B

(g) OGBN-arxiv      (h) Proteins      (i) Pubmed

Figure 7: Signal-to-noise ratio changes with diffusion steps.

## C    Training loss curve

In this section, we analyze the training losses of our directional diffusion model and the vanilla diffusion approach, as shown in Figure 8. Apparently, our model converges faster.

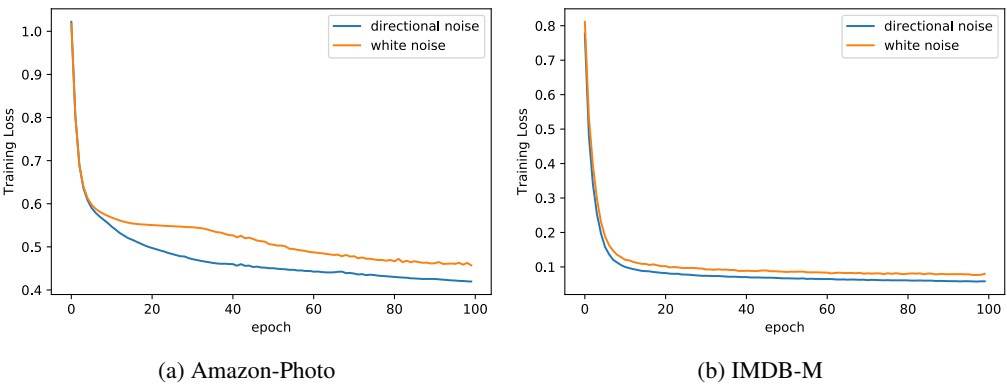

(a) Amazon-Photo      (b) IMDB-M

Figure 8: The training loss curve.

## D   Ablation studies

Table 5 collects the network structures of the baselines used in the comparison experiments below. Specifically, existing approaches, including contrastive learning, GraphMAE, and our proposed method, utilize similar foundational architectures and have comparable number of parameters.

Table 5: Comparing architectures.

| Method | GCN | MLP | Number of GCN Layers |
|--------|-----|-----|----------------------|
| GraphMAE | ✓ | ✓ | 2-4 |
| MVGRL | ✓ | ✓ | 4 |
| Our (DDM) | ✓ | ✓ | 4 |

We also perform additional ablation experiments to evaluate the impact of our specific denoising architecture designs, such as symmetric skip-connections and symmetric network structures. Table 6 collects the results (classification accuracy), which validate our choice to incorporate U-net-inspired ideas and demonstrate the effectiveness of these detailed design choices.

Table 6: An ablation study on the architectured design.

| | Citeseer | PubMed | MUTAG |
|--|----------|--------|-------|
| wo-head | 73.1±0.2 | 80.2±0.2 | 87.8±1.4 |
| wo-encoder | 73.4±0.1 | 81.4±0.3 | 88.9±1.3 |
| wo-skip_connection | 73.5±0.2 | 81.3±0.5 | 86.7±1.1 |
| Baseline | 74.3±0.3 | 81.7±0.8 | 91.51±1.4 |

## E   The complete algorithm

In this section, we present the complete algorithm for our proposed directional diffusion models.

---

**Algorithm 1** The training algorithm.

---

**Input:** A batch of graphs $\mathcal{G} = \{G_1, \cdots G_B\}$
**Output:** The denoising network $f_\theta$

1: **Initialize**: the denoising network $f_\theta$
2: **Compute** $\mu$, the mean of node features across batch $\mathcal{G}$
3: **Compute** $\sigma$, the standard deviation of node features across batch $\mathcal{G}$
4: **while** not convergence **do**
5:   **for** $G_i$ in $\mathcal{G}$ **do**
6:     **for** $t = 1, \ldots, T$ **do**
7:       **Sample** directional noise $\epsilon'$ using equation (2)
8:       **Take** gradient descent step on
          $\nabla_\theta \left\| \mathbf{X}_0 - f_\theta(\sqrt{\bar{\alpha}_t}\mathbf{X_i} + \sqrt{1 - \bar{\alpha}_t}\epsilon', \mathbf{A}, t) \right\|$
9:     **end for**
10:   **end for**
11: **end while**

---

**Algorithm 2** Extracting representations.

**Input:** $G = (\mathbf{A}, \mathbf{X})$, forward step set $\{T_0, T_1, \ldots, T_K\}$, pre-trained denoising network $f_\theta$
**Output:** $\mathbf{H}$, the representation of $G$

1: **Compute** $\mu$ the mean of node features
2: **Compute** $\sigma$ the standard deviation of node features
3: **for** $k$ in $\{T_0, T_1, \ldots, T_K\}$ **do**
4:     **Sample** directional noise $\epsilon'$ using equation (2)
5:     $\mathbf{X_k} \leftarrow \sqrt{\bar{\alpha}_k}\mathbf{X_0} + \sqrt{1 - \bar{\alpha}_k}\epsilon'$
6:     $\mathbf{H}_k \leftarrow f_\theta(\mathbf{X_k}, \mathbf{A}, k)$
7: **end for**
8: **Concatenate** $\mathbf{H} = [\mathbf{H}_{T_0}, \mathbf{H}_{T_1}, \ldots, \mathbf{H}_{T_K}]$
9: **return** $\mathbf{H}$

# F   Statistics and hyper-parameters

In this section, we provide the statistics and hyperparameters in the main experiments in Table 7 and Table 8. The description of each hyperparameter is collected in Table 9.

Table 7: Statistics and hyper-parameters for node classification datasets. "s" indicates multi-class classification, and "m" indicates multi-label classification.

|  | Dataset | Cora | Citeseer | PubMed | Ogbn-arxiv | Computer | Photo |
|---|---|---|---|---|---|---|---|
| Statistics | # nodes | 2708 | 3327 | 19717 | 169343 | 13752 | 7650 |
|  | # edges | 5429 | 4732 | 44338 | 1166243 | 245861 | 119081 |
|  | # classes | 7(s) | 6(s) | 3(s) | 40(s) | 10 | 8 |
| Hyper-para. | feat_drop | 0.1 | 0.2 | 0.2 | 0.2 | 0.4 | 0.2 |
|  | attn_drop | 0.3 | 0.4 | 0.4 | 0.2 | 0.2 | 0.3 |
|  | num_head | 4 | 4 | 4 | 4 | 4 | 4 |
|  | num_hidden | 1024 | 1024 | 1024 | 512 | 512 | 1024 |
|  | learning_rate | 6e-5 | 2e-4 | 2e-4 | 2e-4 | 2e-4 | 1e-4 |
|  | norm | LayerNorm | LayerNorm | LayerNorm | LayerNorm | BatchNorm | BatchNorm |
|  | beta_schedule | Sigmoid | Linear | Const | Linear | Quad | Sigmoid |

Table 8: Statistics and hyper-parameters for graph classification datasets.

|  | Dataset | IMDB-B | IMDB-M | COLLAB | REDDIT-B | PROTEINS | MUTAG |
|---|---|---|---|---|---|---|---|
| Statistics | # graphs | 1000 | 1500 | 5000 | 2000 | 1113 | 188 |
|  | # classes | 2 | 3 | 3 | 2 | 3 | 2 |
|  | Avg. # nodes | 19.8 | 13.0 | 74.5 | 429.7 | 13.0 | 17.9 |
| Hyper-para. | feat_drop | 0.4 | 0.4 | 0.4 | 0.2 | 0.2 | 0.2 |
|  | attn_drop | 0.4 | 0.4 | 0.4 | 0.4 | 0.2 | 0.1 |
|  | num_head | 2 | 4 | 4 | 8 | 4 | 4 |
|  | num_hidden | 128 | 512 | 512 | 512 | 512 | 512 |
|  | learning_rate | 1e-5 | 1e-5 | 1e-5 | 3e-4 | 3e-4 | 3e-4 |
|  | norm | LayerNorm | LayerNorm | LayerNorm | LayerNorm | LayerNorm | LayerNorm |
|  | beta_schedule | Sigmoid | Linear | Const | Linear | Linear | Sigmoid |

Table 9: Hyper-parameter description.

| Hyper-parameter | Interpretation |
| --- | --- |
| feat_drop | the drop-out rate of hidden layers |
| attn_drop | the drop-out rate of attention modules |
| num_head | the number of heads |
| num_hidden | the number of hidden layers |
| learning_rate | the learning rate in training stage |
| norm | the method of normalization |
| beta_schedule | the schedule of $\beta_t$ |