# OpenReview forum: "Directional diffusion models for graph representation learning"
_NeurIPS.cc/2023/Conference — NeurIPS 2023 poster_

### Official Review · Reviewer_ejpR · 2023-06-28

**Soundness:** 4 excellent
**Presentation:** 2 fair
**Contribution:** 4 excellent
**Rating:** 7
**Confidence:** 2

**Summary:**

The authors present directional diffusion model (DDM) to learn graph / node representation. Compared to vanilla diffusion based representation learning techniques, DDM adds a batch based mean and variance, as well as preserving direction of diffusion. The authors then demonstrate that their model perform better than other models on representating learning by experimenting on various dataset and compare with other methods.

**Strengths:**

- It is a nice finding by the authors that compared to diffusion model for generative modeling, diffusion model for representation learning does not require sampling from final distribution, thus we don't need to know the limiting behavior of the diffusion process.
- The experiments supports the authors' claim on the performance of the model.

**Weaknesses:**

- The authors should include a brief introduction on how diffusion models work on representation learning

**Questions:**

- A lot of citations by the paper, including some baselines like Infograph and GraphMAE, come from arxiv. Are these methods reliable?
- Directional diffusion preserves direction, so in recovery process, won't the representation fail to learn the direction of the entries? If it doesn't learn the directions, does directions don't matter?

---

> ### Author Rebuttal · Authors · 2023-08-09
>
>
> > A lot of citations by the paper, including some baselines like Infograph and GraphMAE, come from arxiv. Are these methods reliable?
> >
> We apologize for our problem, in section 5, the articles we compared and cited are all published in top-tier conferences, as follows： GIN-ICLR 201, DiffPool-NIPS 2018, GCC-KDD 2020, JOAO-ICML 2021, GraphMAE-KDD 2022, MVGRL-ICML 2022, GPT-GNN-KDD 2022, DGI-ICLR 2019, GRACE-ICML 2020, BGRL-ICLR 2022, InfoGCL - NIPS 2021, CCS-SSG nips 2021
> We will correct the problem of citing the arxiv version of GIN, GraphMAE, GARGE, BGRL in the subsequent version.
>
> > Directional diffusion preserves direction, so in recovery process, won't the representation fail to learn the direction of the entries? If it doesn't learn the directions, does directions don't matter?
> >
> In this paper, $\mu$ and $\sigma$ in section 3 are calculated in mini-batch during the model training stage. The core idea of this design is that, through random batch learning of multiple epochs, the model can both approximate the global (law of large numbers) and retain the signal-to-noise ratio in each learning process.
>
> > The author should include an introduction on how diffusion models work on representation learning.
> >
>
> Thanks for your suggestions. Actually, Section 1, the introduction part of our article has discussed some notable and recent works about how diffusion models work in representation learning.  We would like to provide a thorough summary in this regard.
>
> Several methods based on diffusion models[1,2,3,4] have been proposed for effective representation learning. Notably, [4] have demonstrated the value of intermediate activations obtained from denoising networks, as they contain valuable semantic information that can be utilized for tasks like image representation and semantic segmentation. Their findings emphasize the effectiveness of diffusion models in learning meaningful visual representations. More recently, [5] has revealed that the restoration of data corrupted with specific noise levels provides an appropriate pretext task for the model to learn intricate visual concepts, and prioritizing such noise levels over other levels during training improves the performance of diffusion models.
>
> [1] Zhang Z, Zhao Z, Lin Z. Unsupervised representation learning from pre-trained diffusion probabilistic models[J]. Advances in Neural Information Processing Systems, 2022, 35: 22117-22130.
>
> [2] Preechakul K, Chatthee N, Wizadwongsa S, et al. Diffusion autoencoders: Toward a meaningful and decodable representation[C]//Proceedings of the IEEE/CVF Conference on Computer Vision and Pattern Recognition. 2022: 10619-10629.
>
> [3] Abstreiter K, Mittal S, Bauer S, et al. Diffusion-based representation learning[J]. arXiv preprint arXiv:2105.14257, 2021.
>
> [4] Baranchuk D, Rubachev I, Voynov A, et al. Label-efficient semantic segmentation with diffusion models[J]. arXiv preprint arXiv:2112.03126, 2021.
>
> [5] Choi J, Lee J, Shin C, et al. Perception prioritized training of diffusion models[C]//Proceedings of the IEEE/CVF Conference on Computer Vision and Pattern Recognition. 2022: 11472-11481.

---

> > ### Author Response · Authors · 2023-08-20
> >
> > Thank you very much for your questions. If you have any remaining concerns regarding our article, we cordially welcome you to present them, and we shall exert our utmost efforts to address them comprehensively.

---

### Official Review · Reviewer_8dwQ · 2023-07-03

**Soundness:** 3 good
**Presentation:** 2 fair
**Contribution:** 4 excellent
**Rating:** 6
**Confidence:** 4

**Summary:**

This paper offers a good study on anisotropic and directional structure learning of the diffusion model. The authors first conduct a statistical analysis emphasizing the importance of anisotropic structure in graph dataset and demonstrate the disadvantage of the vanilla diffusion model through signal-to-noise ratio. Later on, they propose their work, which is a new pipeline considering preserving the characteristic of data in the diffusion process. Its idea is simple: adding directional diffusion on data, preserving more information than vanilla diffusion. Its model is simple, leveraging 4 layers GNNs and one layer MLP, it achieves improvement in several benchmarks. The idea is elegant.

**Strengths:**

Good idea! The work not only offers inspiration to the graph learning community, but it also contributes to the diffusion model. The two constraints added in the work are elegant and proper.

**Weaknesses:**

The writing is not so well, especially the formula part. There are too many mistakes there. Their performances aren't satisfying, but I believe the underlying problem doesn't come from the pipeline, it comes from the simple model. There aren't convergence proofs for their learning architecture. Noticed that the diffusion model, which stems from the quasi-static process, possesses a good property of converging. I hope to see proof of the effectiveness of the constraints adding to the original process.

**Questions:**

1. Do you have proof of the effectiveness of your idea?
2. Can you offer more experimental results?

**Limitations:**

1. They haven't provided limitations for their work.
2. Though the idea is inspirational, the writing is not good, and there are too many writing mistake.

---

> ### Author Rebuttal · Authors · 2023-08-09
>
> > Do you have proof of the effectiveness of your idea?
> >
>
> Indeed, in this paper, we didn’t provide some theoretical proof. However, we believe that the problem studied in this paper and our contributions are considerably interesting and extensible. Just as mentioned by Reviewer 5, for the graph learning community, our research introduces a novel paradigm for graph representation learning and provides comprehensive experimental evidence of the effectiveness of this paradigm. For the diffusion model community, our study also demonstrates impacts on specific data distributions with diffusion models and we propose a data-dependent forward process. And thus, as a pioneering work studying the impacts of data and focusing on data-dependent cases, it is difficult to formulate our method into existing theoretical frameworks. We are actively working on the theoretical proof and believe that it will be addressed in the future.
>
> > Can you offer more experimental results?
> >
> Extensive results on baselines (2 tasks and 12 datasets) demonstrate that this method provides a novel solution for graph representation learning. Apart from contrast experiments, we added an ablation experiment on the model structure:
> |          | GCN | MLP | Number layers of GCN |
> |----------|-----|-----|----------------------|
> | GraphMAE | ✓   | ✓   | 2-4                  |
> | MVGRL    | ✓   | ✓   | 4                    |
> | Our(DDM) | ✓   | ✓   | 4                    |
>
> |  | Citeseer | PubMed | MUTAG |
> | --- | --- | --- | --- |
> | wo-head | 73.1±0.2 | 80.2±0.2 | 87.8±1.4 |
> | wo-encoder | 73.4±0.1 | 81.4±0.3 | 88.9±1.3 |
> | wo-skip_connection | 73.5±0.2 | 81.3±0.5 | 86.7±1.1 |
> | Baseline | 74.3±0.3 | 81.7±0.8 | 91.51 ±1.4 |
>
> it can be seen that considering the similar parameter amount and modules with the baseline, our introduction of diffusion pre-training such as the U-net structure plays an important role.
> Moreover, as we discussed with review 2 by quantifying the anisotropy of data, we show that our method can achieve better performance on datasets with strong anisotropy

---

> > ### Comment · Reviewer_8dwQ · 2023-08-12
> > **Response to the authors' rebuttal.**
> >
> > Thanks for offering kind responses to my questions. I hope you could correct writing mistakes and make the paper more readable. Please offer a elegant theorectical proof as soon as possible, the idea of the paper is innovative and good.

---

### Official Review · Reviewer_tkhv · 2023-07-06

**Soundness:** 2 fair
**Presentation:** 2 fair
**Contribution:** 2 fair
**Rating:** 5
**Confidence:** 5

**Summary:**

This paper presents a method named Directional Diffusion Model (DDM) for unsupervised representation learning, targeting applications in graph and node classification. The model's performance is evaluated on various benchmark datasets and compared to both unsupervised and supervised models. The results demonstrate that DDM can effectively learn meaningful graph- or node-level representations. Furthermore, the paper offers some exploration of how different types of noise impact the learning process.

**Strengths:**

1. The authors present empirical evaluations of DDM on multiple benchmark datasets, which validate the effectiveness of the proposed method to some extent.

2. The paper includes an investigation into the impact of different types of noise, which adds depth to the analysis and understanding of the proposed method.

**Weaknesses:**

1. The SVD visualizations could be more insightful. It's observed that the 2D projections computed from graph datasets appear biased and are predominantly on the right plane. However, the methodology behind these visualizations remains unclear. Are they based on singular values? Further explanation would be beneficial.

2.  The Signal-to-Noise Ratio (SNR) plots in the supplementary material show comparable results under white noise and directional noise. This implies that the DDM may not offer significant advantages on these datasets. The authors should provide some clarification.

3. The paper is primarily empirical and lacks theoretical foundations, which might limit its generalizability.

4.  The algorithm snippet doesn't seem to provide sufficient implementation details, which might impede attempts to reproduce the study.

**Questions:**

1. Please clarify the issues raised in the weaknesses section.

2. How does DDM compare with widely-known baselines in terms of performance, and are there any specific scenarios where DDM particularly excels or falls short?

3. Regarding the means and deviations at line 170: are these calculated relative to the node features at time 0, or do they consider other time steps?


In its current form, I believe the paper does not meet the rigorous quality and contribution standards expected at a top-tier conference like NeurIPS.

**Limitations:**

The authors did not explicitly state any limitations of their study. However, the identified weaknesses could be viewed as potential limitations. These include the lack of theoretical foundations and potential issues in reproducibility due to insufficient implementation details. Additionally, the paper could benefit from a clear discussion of the limitations of DDM in handling specific scenarios or types of data.

---

> ### Author Rebuttal · Authors · 2023-08-09
>
> > The SVD visualizations could be more insightful. However, the methodology behind these visualizations remains unclear.
> >
> Thanks for your suggestions,  Using the SVD decomposition to analyze the anisotropic is first proposed by [1] and has been widely used in NLP[1,2,3,4,5]. Specifically, we calculate the SVD decomposition of the graphs’ node feature matrix and then choose the first and second singular vectors as the x-axis and y-axis.  The methodology behind these visualizations, as discussed in section 1, line 44, is to gain an intuitive understanding of the differences in distribution between graph data and image data. Prompted by this, we further examine the disadvantage of the vanilla diffusion model through the signal-to-noise ratio under such data and propose our new pipeline considering preserving the characteristic of data in the diffusion process.
> From a singular value perspective, we can draw similar conclusions about anisotropy. This method was also introduced in [1].  We follow the method in [1] to measure the anisotropy of the data by measuring the ratio of the sum of the first two singular values to the sum of the first ten singular values. The results are as follows,
>
> > The Signal-to-Noise Ratio (SNR) plots in the supplementary material show comparable results under white noise and directional noise.....
> >
> We follow the approach of [1] to measure the anisotropy of the data by measuring the ratio of the sum of the first two singular values to the sum of the first ten singular values. The results are as follows:
>
> |  | Top@2/Top@5 |
> | --- | --- |
> | ogbn-arxiv | 0.460 |
> | citseer | 0.532 |
> | Pubmed | 0.533 |
> | Computer | 0.722 |
> | Cora | 0.419 |
>
> We can observe that on datasets with low anisotropy (Ogbn-arxiv, cora), our approach performs comparably to previous methods and falls short of supervised models. On datasets with strong anisotropy (Pubmed, Citeseer, Computer), our model has made significant strides, even surpassing some supervised training outcomes. As we discussed in section 5.1 & section .5.2  By utilizing the directional noise diffusion, our method acts as a pseudo-infinite-step data augmentation technique that generates numerous samples while preserving data structure.
> > The paper is primarily empirical and lacks theoretical foundations, which might limit its generalizability.
> >
> Indeed, in this paper, we didn’t provide some theoretical proof. But we believe that the problem studied in this paper and our contributions are considerably interesting and extensible. Just as mentioned by Reviewer 5, for the graph learning community, our research introduces a novel paradigm for graph representation learning and provides comprehensive experimental evidence of the effectiveness of this paradigm. For the diffusion model community, our study also demonstrates impacts on specific data distributions with diffusion models and we propose a data-dependent forward process. And thus, as a pioneering work studying the impacts of data and focusing on data-dependent cases, it is difficult to formulate our method into existing theoretical frameworks. We are actively working on the theoretical proof and believe that it will be addressed in the future.
>
> > The algorithm snippet doesn't seem to provide sufficient implementation details, which might impede attempts to reproduce the study.
> >
>
> We apologize for not providing the definitions of $X_i$ and $A$ in the algorithm in Appendix: as we have defined in the paper, $X_i$ is the feature matrix of graph $\mathcal{G}_i$， while $A$ is the adjacency matrix of $\mathcal{G}$. Every time taking the gradient step, the parameters $\theta$ are renewed.
>
> > How does DDM compare with widely-known baselines in terms of performance, and are there any specific scenarios where DDM particularly excels or falls short?
> >
>
> Actually, we have conducted extensive experiments in section 5.1 and section 5.2 to compare our models with baselines, and the results are shown in Table 1 and Table 2: in graph classification tasks, we surpass the performance of all baselines on IMDB-B, IMDB-M, COLLAB, PROTEINS, and MUTAG datasets; in node classification tasks, we outperform those baselines on Citesser, PubMed, Amazon-Computer and Amazon-Photo datasets. The falling short and excelling performance are also discussed in section 5, which are attributed to the isotropy of data in short.
>
> > Regarding the means and deviations at line 170: are these calculated relative to the node features at time 0, or do they consider other time steps?
> >
>
> As mentioned in line 171, $\mu$ and $sigma$ are the means are deviations of the raw features vectors at time 0, and they can alleviate the impacts resulting from the rapid decline of SNR as discussed in section
>
> [1] Gao J, He D, Tan X, et al. Representation degeneration problem in training natural language generation models[J]. arXiv preprint arXiv:1907.12009, 2019.
>
> [2] Qiu R, Huang Z, Yin H, et al. Contrastive learning for representation degeneration problem in sequential recommendation[C]//Proceedings of the fifteenth ACM international conference on web search and data mining. 2022: 813-823.
>
> [3] Zou D, Wei W, Mao X L, et al. Multi-level cross-view contrastive learning for knowledge-aware recommender system[C]//Proceedings of the 45th International ACM SIGIR Conference on Research and Development in Information Retrieval. 2022: 1358-1368.
>
> [4] Wang L, Huang J, Huang K, et al. Improving neural language generation with spectrum control[C]//International Conference on Learning Representations. 2019.
>
> [5] Yu S, Song J, Kim H, et al. Rare Tokens Degenerate All Tokens: Improving Neural Text Generation via Adaptive Gradient Gating for Rare Token Embeddings[C]//Proceedings of the 60th Annual Meeting of the Association for Computational Linguistics (Volume 1: Long Papers). 2022: 29-45.

---

> > ### Comment · Reviewer_tkhv · 2023-08-11
> > **thank you for the responses**
> >
> > The feedback from the reviewers has addressed some of my concerns, prompting me to increase the score to 5.

---

> > > ### Author Response · Authors · 2023-08-11
> > >
> > > Thank you very much for your active engagement. If you have any remaining concerns regarding our article, we cordially welcome you to present them, and we shall exert our utmost efforts to address them comprehensively.

---

### Official Review · Reviewer_jRHd · 2023-07-07

**Soundness:** 3 good
**Presentation:** 3 good
**Contribution:** 3 good
**Rating:** 5
**Confidence:** 3

**Summary:**

This study proposed adding directional noise on learning anisotropic graphs using diffusion models. The study adds new perspectives in exploring the anisotropic structures in graph data. The numerical results are promising to support the authors' ideas.

**Strengths:**

I find this is an interesting paper - the rational is convincing, and the authors performed extensive experiments to support the idea. The discussions on the noise and ablation studies provide further insights into understanding the usefulness of adding directional noises. The paper provides a valuable perspective in understanding diffusion models for graph learning.

**Weaknesses:**

The authors mainly prove the utility of the proposed approach through experiments; there is a lack of theorectical proof.

**Questions:**

The authors have performed extensive experiments, but it would add more evidence if the authors can add some theoretical proof.

For figure 3, does each dot represent a graph? Are they from simulated datasets?


**Limitations:**

The authors have discussed several future directions that can be used to improve the current model.

---

> ### Author Rebuttal · Authors · 2023-08-09
>
> > For figure 3, does each dot represent a graph? Are they from simulated datasets?
> >
>
> Thanks for your question, Figure 3 shows a set of simulated data. Each sample point comes from an anisotropic (covariance is not equal to the identity matrix) normal distribution. As discussed in section 4, the simulation is designed to show that when the data distribution is anisotropy, our proposed directional can better preserve the structural information of the data during the forward process.
>
> > The authors have performed extensive experiments, but it would add more evidence if the authors can add some theoretical proof.
> >
>
> Thanks for your advice! Indeed, in this paper, we didn’t provide some theoretical proof. However, we believe that the problem studied in this paper and our contributions are considerably interesting and extensible. Just as mentioned by Reviewer 5, for the graph learning community, our research introduces a novel paradigm for graph representation learning and provides comprehensive experimental evidence of the effectiveness of this paradigm. For the diffusion model community, our study also demonstrates impacts on specific data distributions with diffusion models and we propose a data-dependent forward process. And thus, as a pioneering work studying the impacts of data and focusing on data-dependent cases, it is difficult to formulate our method into existing theoretical frameworks. We are actively working on the theoretical proof and believe that it will be addressed in the future.

---

> > ### Comment · Reviewer_jRHd · 2023-08-15
> >
> > I appreciate the authors' responses. I don't have any other questions.

---

### Official Review · Reviewer_BUxE · 2023-07-24

**Soundness:** 2 fair
**Presentation:** 3 good
**Contribution:** 2 fair
**Rating:** 6
**Confidence:** 3

**Summary:**

This work proposes a class of diffusion models to improve the accuracy of graph representation learning. The model incorporates both data-dependent and anisotropic noise in the forward noising process, by scaling its magnitude and direction based on each coordinate of the data. This structured noise maintains the signal present in the data over longer time windows than using standard isotropic white noise during the forward noising process. The authors show that this model improves upon state-of-the-art methods on a large collection of benchmark datasets. Moreover, they perform an ablation study to understand the effect of the two proposed modification to the noise process.

**Strengths:**

- The authors visually explain and empirically demonstrate the effect of using non-isotropic noise in the forward diffusion process to improve classification tasks. These noise processes provide clear intuition why this is preferred to white noise for these tasks.
- The authors propose a novel strategy for extracting graph representations based on time-dependent denoiser that combines graph neural networks and UNet architectures.
- The application of diffusion models to graph representation learning is novel and the new model is shown to yield superior results to existing algorithms.

**Weaknesses:**

It would be great for the authors to comment and compare with other non-isotropic noise processes that have been proposed for diffusion models for sampling and generative modeling. Some examples outside of the graph representation learning context are:
* Score-based Denoising Diffusion with Non-Isotropic Gaussian Noise Models, Vikram Voleti, Christopher Pal, Adam Oberman, 2022
* Blurring Diffusion Models, Emiel Hoogeboom, Tim Salimans, 2023

In addition to the changes in the noise process, the authors propose a specific architecture for the denoiser, and selection of representative features from their hidden layers. This architecture could be relevant on its own to extract representations of the dataset, without the denoising time components. Do any of the compared methods investigate how this denoiser architecture, without the diffusion model, would perform for representation learning. This might be helpful as an additional ablation study to see the effect of the architecture.

The metrics used to evaluate their experiments can be described in more detail. The values for the results in Tables 1,2,3 were not clearly mentioned in the caption or the main text. Mathematical equations may also be helpful to precisely describe the accuracy measurements in Figure 5.

The authors comment that the word vectors often exhibit greater anisotropy, which yields superior performance in node classification tasks. It would be great if the authors could quantify this to validate this claim that the performance improves with more anisotropy.

**Questions:**

An important choice for defining the representations are the user selected time-steps where the outputs are extracted from the denoising network. Do the authors have a guideline or recommendation (e.g., based on the empirical studies) for how to choose these time steps. This seems particularly relevant given that the SNR does not degrade monotonically for some datasets (e.g., IMDB in Figure 2).

Even if the SNR in standard diffusion models decay quickly with increasing time, how do the results presented here compare to a baseline diffusion model with isotropic noise where the time-steps are chosen "optimally". For example, what if the time-steps are chosen all near the initial time, $t = 0$, when there is still signal contained in the noisy data?

**Limitations:**

The authors show improved results on almost all datasets. It would be great if the authors could comment on what leads to the similar performance on the Ogbn-arxiv dataset. Is it because the data is more isotropic in this case? When do the authors expect the algorithm to under-perform for the evaluated tasks?

In addition to the questions above, it would be good to address these minor comments:
- Define the acronym for their proposed framework, DDM
- Include standard errors for the results in Table 3
- Explain why $f_\theta$ depends on $A$ in equation (4). It is sufficient to highlight that this is the structure of a GNN, which may be helpful for some readers.
- Clarify in Figure 4 that the directional noise is only added to $X_0$ and not $A$.

---

> ### Author Rebuttal · Authors · 2023-08-09
>
> > It would be great for the authors to comment and compare with other non-isotropic noise processes
> >
> With full respect, The existing literature you mentioned differs fundamentally from our approach. As pointed out by reviewer 6, our paper proposes that the ultimate distribution of the diffusion process isn't essential; what's crucial is the rate of SNR attenuation. Thus, the noise we propose is data-dependent, with its distribution determined by the mini-batch data.(Section 4  eq2 & eq3) .The main objective of this design is to curb the speed of SNR reduction. In contrast, existing literature still relies on methods that are non-data-dependent, using fixed noise distributions.
>
> > This might be helpful as an additional ablation study to see the effect of the architecture.
> >
> We sort out the network structures of the baselines that appeared in the comparison experiments below. Existing schemes, including contrastive learning, GrapMAE, and our proposed method use similar basic structures (gat) and amounts of parameters, which shows that our proposed unsupervised representation learning based on diffusion models, plays a dominant role in improving the representation. In addition, we conduct further ablation experiments for our special designs in the denoiser architecture, including symmetric skip-connection, and symmetric network structure. The results are as follows, which prove the correctness of choosing to transplant the U-net idea and the effectiveness of the detailed design.
> |  | GCN | MLP | Number layers of GCN |
> | --- | --- | --- | --- |
> | GraphMAE | ✓ | ✓ | 2-4 |
> | MVGRL | ✓ | ✓ | 4 |
> | Our(DDM) | ✓ | ✓ | 4 |
>
> |  | Citeseer | PubMed | MUTAG |
> | --- | --- | --- | --- |
> | wo-head | 73.1±0.2 | 80.2±0.2 | 87.8±1.4 |
> | wo-encoder | 73.4±0.1 | 81.4±0.3 | 88.9±1.3 |
> | wo-skip_connection | 73.5±0.2 | 81.3±0.5 | 86.7±1.1 |
> | Baseline | 74.3±0.3 | 81.7±0.8 | 91.51 ±1.4 |
>
> > The metrics used to evaluate their experiments can be described in more detail.
> >
> We explained the evaluation metrics used in Sections 5.2 and 5.3. For node classification tasks, we report the classification accuracy of nodes in the validation set. And for graph classification tasks, we report the accuracy of the validation set. We will add this to the caption.
>
> > The authors comment that the word vectors often exhibit greater anisotropy, ...... It would be great if the authors could quantify this to validate the claim that the performance improves with more anisotropy.
> The authors show improved results on almost all datasets. It would be great if the authors could comment on what leads to the similar performance on the Ogbn-arxiv dataset. (Question 1)
> >
> We follow the method in [1] to measure the anisotropy of the data by measuring the ratio of the sum of the first two singular values to the sum of the first five singular values. The results are as follows:
>
> |  | Top@2/Top@5 |
> | --- | --- |
> | ogbn-arxiv | 0.460 |
> | citseer | 0.532 |
> | Pubmed | 0.533 |
> | Computer | 0.722 |
> | Cora | 0.419 |
>
> We can observe that on datasets with low anisotropy (ogbn-arxiv, cora), our approach performs comparably to previous methods but falls short of supervised models. On datasets with strong anisotropy (Pubmed, Citeseer, Computer), our model has made significant strides, even surpassing some supervised training outcomes. This indicates that our approach can better learn the specific structure of such graph data.
>
> > An important choice for defining the representations are the user selected time-steps where the outputs are extracted from the denoising network. Do the authors have a guideline or recommendation (e.g., based on the empirical studies).....
> >
>
> Thank you for your inquiry. We are pleased to offer our guidelines in this regard.
> Generally speaking, the signal-to-noise ratio indicates important information in both the training and inference stages. From fig5 in the main text and the fig2 in the appendix, we can see that the trend of SNR is almost proportional to the accuracy of the test set. For example, when T-->1000, SNR of Citseer 1.5-->0.5, acc 0.741-->0.695. Additionally, considering that our final metric surpasses the performance of each individual timestep. we suggest that the selected multi time-step can cover different intervals of the SNR curve to get the best results.
>
> > Include standard errors for the results in Table 3
> >
> We have updated the new table with the standard deviations below.
>
>
> | DataSet | w/o S&R    | w/o R      | Full       |
> |---------|------------|------------|------------|
> | Citseer | 34.37±0.5  | 60.77±0.2  | 74.3 ± 0.3 |
> | PubMed  | 73.07±0.7  | 77.60±0.4  | 81.7 ± 0.8 |
> | IMDB-M  | 49.80±0.53 | 50.87±0.49 | 52.53±0.31 |
> | COLLAB  | 80.50±0.36 | 81.04±0.17 | 81.72±0.31 |
> | MUTAG   | 82.89±1.16 | 87.25±1.12 | 91.51±1.45 |
>
>
> > Explain why $f_\theta$ depends on $A$ in equation (4). It is sufficient to highlight that this is the structure of a GNN, which may be helpful for some readers.
> >
> In this paper, we parameterize $f_\theta$ as GCN. We will correct this unclear expression in the subsequent version.
> > Clarify in Figure 4 that the directional noise is only added to $A_{0}$ and not $A$.
> >
> Thank you for your review, noise is indeed only added to x, and the specific form is in equation (1), (2), (3).
>
> [1] Gao J, He D, Tan X, et al. Representation degeneration problem in training natural language generation models[J]. arXiv preprint arXiv:1907.12009, 2019.

---

> > ### Comment · Reviewer_BUxE · 2023-08-19
> >
> > We thank the authors for their detailed response! The feedback from the reviewers has addressed some of my concerns, so I've raised my score. I would still appreciate some comparisons with non-isotropic noise or with choosing different schedules for the noise as baselines for the proposed data-dependent noise (in the absence of theoretical results on the method).

---

> > > ### Author Response · Authors · 2023-08-20
> > >
> > > Thank you very much for your active engagement.
> > > We are willing to provide more comparisons with non-isotropic noise or with choosing different schedules here ( Same as what we provided to Reviewer 1）.
> > > | Noise schedule function  | Noise type  | Citeseer | PubMed | MUTAG |
> > > |--------------------------|-------------|----------|--------|-------|
> > > | cosine (s=0,e=1,τ = 1)   | DDM         | 0.715    | 0.824  | 0.867 |
> > > | cosine (s=0,e=1,τ = 1)   | White Noise | 0.371    | 0.453  | 0.692 |
> > > | sigmoid (s=-3,e=3,τ = 1) | DDM         | 0.735    | 0.803  | 0.889 |
> > > | sigmoid (s=-3,e=3,τ = 1) | White Noise | 0.672    | 0.606  | 0.689 |
> > > | sigmoid (s=0, e=3,τ = 1) | DDM         | 0.710    | 0.806  | 0.877 |
> > > | sigmoid (s=0, e=3,τ = 1) | White Noise | 0.581    | 0.434  | 0.691 |
> > >
> > > It shows that. Different schedulers do indeed influence the final effectiveness of the model. However, due to the data-independent nature of white noise, its ultimate performance heavily relies on the hyperparameters of the scheduler. Yet, irrespective of the scheduler employed, our proposed data-dependent anisotropic noise consistently yields superior outcomes. This confirms our impression of anisotropic structures in section 1. We will add this experiment to the paper in the subsequent version.

---

### Official Review · Reviewer_emkS · 2023-07-25

**Soundness:** 3 good
**Presentation:** 4 excellent
**Contribution:** 3 good
**Rating:** 7
**Confidence:** 4

**Summary:**

The authors address the gap in unsupervised graph representation learning by exploring the use of diffusion models. They propose directional diffusion models that incorporate data-dependent, anisotropic, and directional noises in the forward diffusion process to better handle anisotropic structures in graphs. Experiments on publicly available datasets showcase the superiority of their models over state-of-the-art baselines, demonstrating their effectiveness in capturing meaningful graph representations. Overall, the paper presents a compelling approach that contributes to the advancement of unsupervised graph representation learning.

**Strengths:**

1. **Motivation**

    The introduction is well-motivated, providing a thorough explanation of the challenge and task at hand. The authors go beyond textual descriptions and use simple visualizations on both real and synthetic data to demonstrate their points effectively. This approach enhances the clarity and understanding of the presented research, making it accessible to a wider audience.
&nbsp;

2. **Method**

    The authors present a straightforward yet effective solution for incorporating directional noise into node embeddings. This approach effectively addresses the challenge posed by anisotropic structures in various domains, including graphs and potentially text data. Their proposed method demonstrates promising results in handling directional noise and enhancing the quality of node and graph embeddings.
&nbsp;

3. **Architecture**

    I appreciate the authors intention to adapt the well-known and effective U-Net architecture from the image domain to the graph domain. The incorporation of skip connections in the U-Net is particularly relevant for denoising tasks. This thoughtful adaptation enhances the model's ability to handle graph-related denoising effectively.
&nbsp;

4. **Experiments**

    The authors conduct a comprehensive comparison with numerous baselines across multiple datasets. Furthermore, their evaluation settings, which involves 10-fold cross-validation with standard deviation after five runs, are robust and reliable. This rigorous evaluation methodology ensures the validity and statistical significance of their results.

**Weaknesses:**

1. **missing releted work**

    There are existing works in the intersection of graphs and diffusions are missing, contradicting the authors statement "To the best of our knowledge, there have been no works for diffusion-model-based graph representation learning.". Some for example:
    - Niu, Chenhao, et al. "Permutation invariant graph generation via score-based generative modeling." International Conference on Artificial Intelligence and Statistics. PMLR, 2020.
    - Xu, Minkai, et al. "Geodiff: A geometric diffusion model for molecular conformation generation." International Conference on Learning Representations, 2022.
    - Vignac, Clement, et al. "Digress: Discrete denoising diffusion for graph generation." International Conference on Learning Representations, 2023.

    Furthermore, some simple techniques can be applied to create a smoother SNR curves over the different diffusion steps. For instance:
    - Chen, Ting. "On the importance of noise scheduling for diffusion models." arXiv preprint arXiv:2301.10972 (2023).

    The same problem was presented in it over the image domain (on high-resolution images), and the solution was to use different noise schedulers. Why not simply try this trick?
&nbsp;

2. **missing details**

   There are missing details in the paper that makes it hard to fully understand and reproduce the papers results. For example:
    - "µ and σ are calculated using graphs within the batch.” -- what is done during inference? it is EMA over what was been seen in training time? does it handle only batch inference?
    - They paper did not state how exactly the entire graph-level representation is obtained, it is sum/mean of all node representations?
    - Algorithm 2 in appendix, line 6: what is exactly $A_t$? should it be just $A$?

**Questions:**

1. "µ and σ are calculated using graphs within the batch.” -- what is done during inference? why not just pre-calculate those on the entire train split?
2. "allowing the latent code to contain meaningful compressed knowledge while preventing the decoder from degrading into an identical mapping” -- why will that happen that requires a dedicate MLP? what is this different from the simple U-Net used in the image domain?
3. Algorithm 2 in appendix, line 6, what is $A_t$?
4. In some cases, the SNR goes even higher for larger diffusion steps (like in the Computer dataset for instance), why is that?
5. How is the entire graph-level representation obtained?
6. Referring to figure 2, shouldn't the SNR be 0 in the last diffusion steps? It doesn't reach zero as the directional noise keeps them separated, is that a wanted outcome? If so, why?
7. Why not apply just a different noise scheduler? instead of linear, coisne for instance. Similar observation was shown on high-resolution images and different schedulers where proposed [1]. I know it is still isotropic noise, so at least show results as a baseline/ablation.


___

[1] Chen, Ting. "On the importance of noise scheduling for diffusion models." arXiv preprint arXiv:2301.10972 (2023).

**Limitations:**

The authors address the limitations of their work, and support their claims with experiments and ablations.

---

> ### Author Rebuttal · Authors · 2023-08-09
>
>
> > **missing releted work,** here are existing works in the intersection of graphs and diffusions are missing ....
>
> First, with full respect, the three existing works you mentioned do not conflict with our statements. These three works you pointed out are devoted to exploring graph structure generation based on diffusion models, whose core goal is to **generate graph–discrete graph structure**. However, instead of graph structure generation, our research focuses on unsupervised graph representation learning tasks, whose purpose is to **generate node-level or graph-level vector unsupervised learning**. We first pre-train a model without supervision and use it to generate node or graph representations for downstream classification tasks.
>
> > Furthermore, some simple techniques can be applied to create a smoother & Why not apply just a different noise scheduler?Why not apply just a different noise scheduler? ....(Question 7)
> >
> Regarding the impacts of different schedules, We adopt the same experimental setup as described in the article you referenced.  The experimental results are shown in the following table:
>
> | Noise schedule function | Noise type | Citeseer | PubMed | MUTAG |
> | --- | --- | --- | --- | --- |
> | cosine (s=0,e=1,τ = 1) | DDM | 0.715 | 0.824 | 0.867 |
> | cosine (s=0,e=1,τ = 1) | White Noise | 0.371 | 0.453 | 0.692 |
> | sigmoid (s=-3,e=3,τ = 1) | DDM | 0.735 | 0.803 | 0.889 |
> | sigmoid (s=-3,e=3,τ = 1) | White Noise | 0.672 | 0.606 | 0.689 |
> | sigmoid (s=0, e=3,τ = 1) | DDM | 0.710 | 0.806 | 0.877 |
> | sigmoid (s=0, e=3,τ = 1) | White Noise | 0.581 | 0.434 | 0.691 |
>
> Different schedulers do indeed influence the final effectiveness of the model. However, due to the data-independent nature of white noise, its ultimate performance heavily relies on the hyperparameters of the scheduler. Yet, irrespective of the scheduler employed, our proposed data-dependent anisotropic noise consistently yields superior outcomes. This confirms our impression of anisotropic structures in section 1.
> > µ and σ are calculated using graphs within the batch.” -- what is done during inference? ....
> >
> In this paper, $\mu$  and $\sigma$ are also calculated in mini-batch during the model inference stage.  In node classification tasks, the term "mini-batch" refers to the nodes currently included in the computation, a requirement that can be easily fulfilled in node classification tasks. In graph classification tasks, "mini-batch" refers to the graphs currently included in the computation.
> > why not just pre-calculate those on the entire train split
> >
> During model inference, compared to pre-calculate or EMA, calculating the *μ* and *σ* in the mini-batch provides a more effective constraint on the diffusion process within the local neighborhood of the batch. This approach can more efficiently ensure a higher SNR.
> In addition, it is worth noting that in order to prevent inference batch size from affecting, we adopted the same batch size in the GraphMAE [2] experiment to ensure the fairness of the benchmark.
> > why will that happen that requires a dedicate MLP? what is this different from the simple U-Net used in the image domain?
> >
> The MLP in our network is composed of linear – relu – linear, which projects the feature map from the hidden dimension back to the data dimension. This is necessary from a computational point of view. By introducing such a nonlinear structure for dimension conversion, GAT can be restricted to learning in the hidden space. This design has proven effective in many classic self-supervised training methods (simclr, simclrv2, SimSiam, moco). Our statement is a possible explanation for the effectiveness of this design. If this structure, containing nonlinearity, is removed, the latter half of the decoder will have to learn the mapping from the hidden space to the original data distribution, and the former half of the decoder may degenerate into an identity map of the encoder part. We also provide more ablation experiments related to the network structure in the responses to reviewer 2.
> > The paper did not state how exactly the entire graph-level representation is obtained, it is sum/mean of all node representations?How is the entire graph-level representation obtained? (Question 5)
> >
> Yes, graph-level representation is derived from pooling node-level representations, and we employ either max-pooling or mean-pooling. The specific approach remains consistent with that of the GraphMAE [2] baseline.
> > Algorithm 2 in appendix, line 6, what is $A_{t}$
> >
> Thank you for your question and we apologize for our negligence. This is just a typo and it should be $A$. Our proposed DDM is for node features.
> > Referring to figure 2, shouldn't the SNR be 0 in the last diffusion steps?
> >
> This phenomenon is consistent with our design. In section 4, our method transforms the data-independent Gaussian noise into an anisotropic and batch-dependent noise ϵˉ and ensures that noise ϵˉ into the same hyperplane of the feature. This makes $\epsilon$ related to $x_{0}$ so that SNR may not reach 0 at the last diffusion steps .
> > In some cases, the SNR goes even higher for larger diffusion steps....
> >
> We believe that this phenomenon arises from the inherent characteristics of certain graphs data. In fact, introducing slighter noise to node features on some graph data has been a well-established technique to enhance classification accuracy. This concept is elaborated [1], So our approach better leverages this particular characteristic of the data and renders it prominent enough to be observable in the SNR curve.
>
> [1] Sato, R., Yamada, M., & Kashima, H. (2021). Random features strengthen graph neural networks. In *Proceedings of the 2021 SIAM international conference on data mining (SDM)* (pp. 333-341). Society for Industrial and Applied Mathematics.
> [2] Hou, Zhenyu, et al. "Graphmae: Self-supervised masked graph autoencoders." *Proceedings of the 28th ACM SIGKDD Conference on Knowledge Discovery and Data Mining*. 2022.

---

> > ### Comment · Reviewer_emkS · 2023-08-18
> >
> > I appreciate the response from the authors which answered almost all my concerns.
> > The only thing I'm not yet convinced about is why the SNR doesn't reach zero - isn't a graph, for example, with all possible edges have SNR=0? I'd appriciate an explanation.
> > Anyway, I'm upgrading my score to accept, as besides this issue, all my questions where fully answered.

---

> > > ### Author Response · Authors · 2023-08-20
> > >
> > > Thank you very much for your active engagement. In our SNR analysis,  we use the LDA method to quantify the signal-to-noise ratio of the **node features** for the node or graph class labeling (this design aims to be consistent with downstream tasks ). Since our batch-dependent noise is always correlated with $x$, the  $x_{t}$ will always be more Informative than pure white noise, so it is still possible that the SNR doesn't reach zero in the last diffusion steps.
> > > If you have any remaining concerns regarding our article, we cordially welcome you to present them, and we shall exert our utmost efforts to address them comprehensively.

---

### Decision · Program_Chairs · 2023-09-21

**Decision:**

Accept (poster)

**Comment:**

The reviewers originally had some concerns but these have mostly been addressed by the authors in the rebuttal and all reviewers now lean for acceptance. According to the reviewers feedback, please incorporate all the discussion in the updated manuscript, paying special attention to adding more context in the background section and being more precise about the metrics used in the evaluations.